# Mediators of monocyte chemotaxis and matrix remodeling are associated with mortality and pulmonary fibroproliferation in patients with severe COVID-19

Sarah E. Holton[ID][1,2]*, Mallorie Mitchem[2], Hamid Chalian[3], Sudhakar Pipavath[1,3], Eric D. Morrell[1], Pavan K. Bhatraju[1], Jessica A. Hamerman[2], Cate Speake[2], Uma Malhotra[3,4,5], Mark M. Wurfel[1], Steven E. Ziegler[2], Carmen Mikacenic[ID][1,2,5]

1 Division of Pulmonary, Critical Care, and Sleep Medicine, Department of Medicine, University of Washington, Seattle, WA, United States of America, 2 Benaroya Research Institute, Seattle, WA, United States of America, 3 Department of Radiology, University of Washington, Seattle, WA, United States of America, 4 Division of Infectious Diseases, Department of Medicine, University of Washington, Seattle, WA, United States of America, 5 Virginia Mason Franciscan Health, Seattle, WA, United States of America

* seholton@uw.edu

**Data Availability Statement:** All relevant data are within the manuscript and its Supporting Information files.

## Abstract

Acute respiratory distress syndrome (ARDS) has a fibroproliferative phase that may be followed by pulmonary fibrosis. Pulmonary fibrosis following COVID-19 pneumonia has been described at autopsy and following lung transplantation. We hypothesized that protein mediators of tissue remodeling and monocyte chemotaxis are elevated in the plasma and endotracheal aspirates of critically ill patients with COVID-19 who subsequently develop features of pulmonary fibroproliferation. We enrolled COVID-19 patients admitted to the ICU with hypoxemic respiratory failure. ($n = 195$). Plasma was collected within 24h of ICU admission and at 7d. In mechanically ventilated patients, endotracheal aspirates (ETA) were collected. Protein concentrations were measured by immunoassay. We tested for associations between protein concentrations and respiratory outcomes using logistic regression adjusting for age, sex, treatment with steroids, and APACHE III score. In a subset of patients who had CT scans during hospitalization ($n = 75$), we tested for associations between protein concentrations and radiographic features of fibroproliferation. Among the entire cohort, plasma IL-6, TNF-α, CCL2, and Amphiregulin levels were significantly associated with in-hospital mortality. In addition, higher plasma concentrations of CCL2, IL-6, TNF-α, Amphiregulin, and CXCL12 were associated with fewer ventilator-free days. We identified 20/75 patients (26%) with features of fibroproliferation. Within 24h of ICU admission, no measured plasma proteins were associated with a fibroproliferative response. However, when measured 96h-128h after admission, Amphiregulin was elevated in those that developed fibroproliferation. ETAs were not correlated with plasma measurements and did not show any association with mortality, ventilator-free days (VFDs), or fibroproliferative response. This cohort study identifies proteins of tissue remodeling and monocyte recruitment are associated with in-hospital mortality, fewer VFDs, and radiographic fibroproliferative response.

**Funding:** This work was sponsored by the National Institutes of Health via the following grant mechanisms: NIH NHLBI T32 HL007287-42 (SEH) NIH U19 AI42733 (CM) NIH NHLBI K23 HL144916 (EDM) NIH NIAID 3R01AI150178-01S1 (JAH).

**Competing interests:** The authors have declared that no competing interests exist.

Measuring changes in these proteins over time may allow for early identification of patients with severe COVID-19 at risk for fibroproliferation.

## Introduction

The acute respiratory distress syndrome (ARDS) is a highly morbid, often fatal syndrome that affects approximately 23% of patients requiring mechanical ventilation. Pulmonary fibrosis following fibroproliferative ARDS has been described previously [1, 2], and was reported following the SARS epidemic of 2003 [3, 4] and MERS outbreak of 2012 [5]. The COVID-19 pandemic has caused severe respiratory failure in many patients [6–9], with longitudinal studies showing that a proportion of survivors of COVID-19 have ongoing respiratory symptoms [10] and decreased lung function [11], while others have persistent radiographic abnormalities including fibrosis [12–14]. Further investigation is needed to understand why this syndrome has such heterogeneous effects and to identify potential therapies that can be used during future viral pandemics. Patients with COVID-19 have a three-fold longer duration of mechanical ventilation compared to patients with influenza [15], which may be due in part to the development of a fibroproliferative phase after viral pneumonia and respiratory failure in patients with severe disease [16–19].

While an associative link between viral ARDS and fibrotic lung remodeling has been established, the underlying mechanism in humans remains unclear. Unlike other studies of post-ARDS fibrosis in which patients have had multiple etiologies for ARDS, studying patients with COVID-19 ARDS provides a unique opportunity to understand the timeline of lung injury and remodeling. Further, with growing numbers of survivors of COVID-19 we are beginning to understand the long-term sequelae both in the lungs and elsewhere in the body. Recent studies have identified pro-fibrotic macrophage populations in the lungs of patients with COVID-19, potentially from recruited circulating monocytes, consistent with prior studies showing monocyte recruitment is linked to the development of pulmonary fibrosis following injury [16, 20]. Other groups have analyzed transcriptional changes in lungs from patients undergoing lung transplant or at autopsy and identified pro-fibrotic programs that are similar to those in idiopathic pulmonary fibrosis [17–19]. This important work establishes a link between fibrotic remodeling during fibroproliferative ARDS and other fibrotic lung diseases, including post-ARDS fibrosis.

Prior studies have established some proteins identified in the circulation or alveolar space that are associated with fibroproliferative ARDS [21–26]. We hypothesized that proteins previously associated with fibrotic lung disease, including idiopathic pulmonary fibrosis, would be elevated in blood and lung fluid from critically ill patients with SARS-CoV-2 who go on to develop fibroproliferative ARDS determined by chest imaging. We measured these proteins in the plasma and endotracheal aspirates of patients at two points during hospitalization to determine associations with mortality, ventilator-free days (VFDs), and development of fibroproliferation. Some of the results of this study have been previously presented in the form of an abstract [27].

## Results

### Plasma cytokines and markers of matrix remodeling are elevated and associated with poor ICU outcomes

Patients were enrolled from three University of Washington ICUs with suspected COVID-19 as previously described [28–30] (*n* = 366) (**Fig 1**). Only those patients with confirmed COVID-19 (based on a positive SARS-CoV-2 PCR test) were included (*n* = 243). Patients were excluded if they were not hypoxemic at the time of enrollment, as defined by the use of

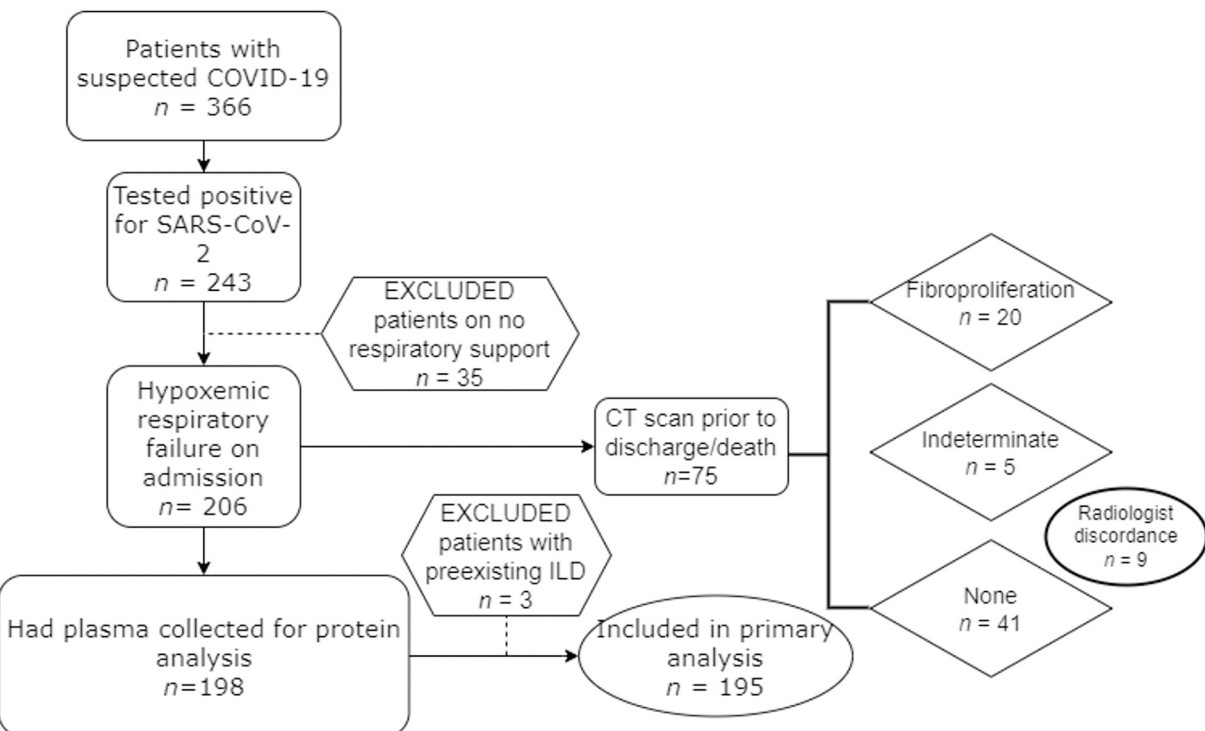

**Fig 1. Development of cohort.** 195 patients were included in the primary outcome analysis. Three patients had pre-existing interstitial lung disease (ILD) from sarcoidosis, idiopathic pulmonary fibrosis, and fibrotic hypersensitivity pneumonitis; they were excluded from analysis.

supplemental oxygen delivered via nasal cannula, high flow nasal cannula, noninvasive ventilation, or invasive mechanical ventilation. A total of 198 patients had plasma collected for protein analysis and three were excluded for having pre-existing interstitial lung disease. Demographics and admission characteristics are shown in **Table 1**. Because the patients were recruited over a long time period, clinical practice regarding vaccination and immune-targeted therapies varied in the cohort. Only 7 patients received a vaccine directed towards SARS-CoV-2. Approximately 80% of the cohort received dexamethasone per protocol as they were recruited after publication of the RECOVERY trial [31].

We hypothesized that proteins previously associated with pulmonary fibrosis would be elevated in the plasma of patients with poor ICU outcomes including increased mortality and fewer ventilator free days (VFDs). We measured proteins listed in **S1 Table in S1 File** and included those that met quality control markers. The cohort had an overall mortality of 43% (**Table 1**). In unadjusted analyses, higher concentrations of IL-6, TNFα, CCL2, Amphiregulin, and CXCL12 measured within 24hrs of ICU admission were associated with in hospital mortality (**S2 and S3 Tables in S1 File**). In models including age, sex, admission APACHE score, and steroid treatment as covariates, doubling of CCL2 (OR 1.3 [CI 1.02–1.73, $p = 0.04$), IL-6 (OR 1.3 [CI 1.12–1.56], $p = 0.0009$), TNF-α (OR 1.75 [CI 1.24–2.6], $p = 0.003$), and Amphiregulin (OR 1.6, [CI 1.1–2.4], $p = 0.019$) remained significantly associated with increased odds of in-hospital mortality (**Fig 2**). When measured in plasma collected at 7 days, only higher concentrations of IL-6 (OR = 2.0, [CI 1.14–5], $p = 0.045$) and Amphiregulin (OR 1.67, [CI 1.1–2.6], $p = 0.02$) were associated with mortality (**S3 Table in S1 File**).

We next studied associations with VFDs. We dichotomized patients to high vs low VFDs based on a threshold of 14 (high ≥14 VFDs, low <14 VFDs) given 60% of the cohort ($n = 118$) had fewer than 14 VFDs (**Table 1**). In unadjusted models, higher concentrations of CCL-2,

**Table 1. Clinical characteristics and selected outcomes of patients within cohort (_n_ = 195) and subset of patients who had CT scans done (_n_ = 75).**

| Characteristics | Cohort (_n_ = 195) | Patients with CT scans (_n_ = 75) |
|---|---|---|
| Age, average (std dev), range, _yr_ | 55.5 (15.4), 20–92 | 54 (13.4), 20–88 |
| Male Sex, _n_ (%) | 133 (68%) | 52 (69%) |
| Ethnicity (% Hispanic) | 126 (65%) | 45 (60%) |
| Race | | |
| • Native American | 5 (2.5%) | 3 (4%) |
| • Pacific Islander | 4 (2%) | 3 (4%) |
| • Asian | 29 (9.8%) | 9 (12%) |
| • Black | 29 (9.8%) | 10 (13%) |
| • White | 115 (59%) | 44 (59%) |
| • Multiracial | 2 (1%) | 2 (2.6%) |
| • Unknown | 11 (5.6%) | 4 (5%) |
| BMI, average, kg/m$^2$ | 31.8 | 32.5 |
| OSH Transfer, _n_ (%) | 101 (52%) | 46 (61%) |
| ARDS at enrollment, _n_ (%) | 74 (38%) | 38 (51%) |
| NIV/HFNC at enrollment, _n_ (%) | 55 (28%) | 21 (28%) |
| Mechanical ventilation at enrollment, _n_ (%) | 100 (51%) | 50 (67%) |
| ECLS at enrollment, _n_ (%) | 16 (8%) | 9 (12%) |
| Received dexamethasone | 147 (75%) | 64 (85%) |
| Received any dose of COVID vaccination, _n_ (%) | 7 (3.5%) | 4 (5%) |
| Admission P:F, median (IQR) | 60 (40–90) | 60 (50–90) |
| APACHE, median (IQR) | 70.5 (52–93.3) | 76 (54–95) |
| RALE score, day 1, median (IQR) | 22 (12–28) | 24 (14–32) |
| COVID test to proximal sample date, median (IQR), _days_ | 4 (1–11) | 7 (1–14) |
| **Outcomes** | | |
| In-hospital death, _n_ (%) | 83 (43%) | 24 (32%) |
| VFDs <14, _n_ (%) | 118 (60%) | 45 (60%) |
| Treatment for refractory hypoxemia, _n_ (%) | 101 (52%) | 41 (55%) |

BMI–body measurement index, OSH = outside hospital, ARDS–acute respiratory distress syndrome, NIV- non-invasive mechanical ventilation, HFNC–high flow nasal cannula, ECLS- extracorporeal life support, COVID-coronavirus disease 2019, P:F–Ratio of the PaO$_2$ to the delivered FiO$_2$, APACHE–Acute Physiology, Age, Chronic Health Evaluation III, RALE–Radiographic Assessment of Lung Edema

IL-6, TNF-α, and Amphiregulin were significantly associated in patients who had few VFDs (**S4 Table in S1 File**). In addition, matrix remodeling proteins MMP-7 and MMP-9 and chemokines P-Selectin and CXCL12 were also elevated in those with fewer VFDs (**S4 Table in S1 File**). When we included these proteins in a model and adjusted for age, sex, admission APACHE score, and steroid treatment, CCL2 (OR .56, [CI 0.38-.78] _p_ = 0.001), IL-6 (OR 0.75, [CI 0.6–0.9], _p_ = 0.0045), TNF-α (OR 0.67, [CI 0.45–0.91], _p_ = 0.03), CXCL12 (OR 0.59, [CI 0.41–0.82], _p_ = 0.0026), and Amphiregulin (OR 0.53, [CI 0.33–0.81], _p_ = 0.005) remained significantly associated with VFDs (**Fig 3**). At the second timepoint, higher plasma concentrations of IL-6, CXCL12, Amphiregulin, MMP7, and P-Selectin remain associated with VFDs only in unadjusted models (**S5 Table in S1 File**).

Because of the nature of our study, patients were enrolled into the ICU at varying points following initial positive SARS-CoV-2 PCR test. To address whether changes in plasma proteins were due to patients being in different states of their disease course, we assessed the association

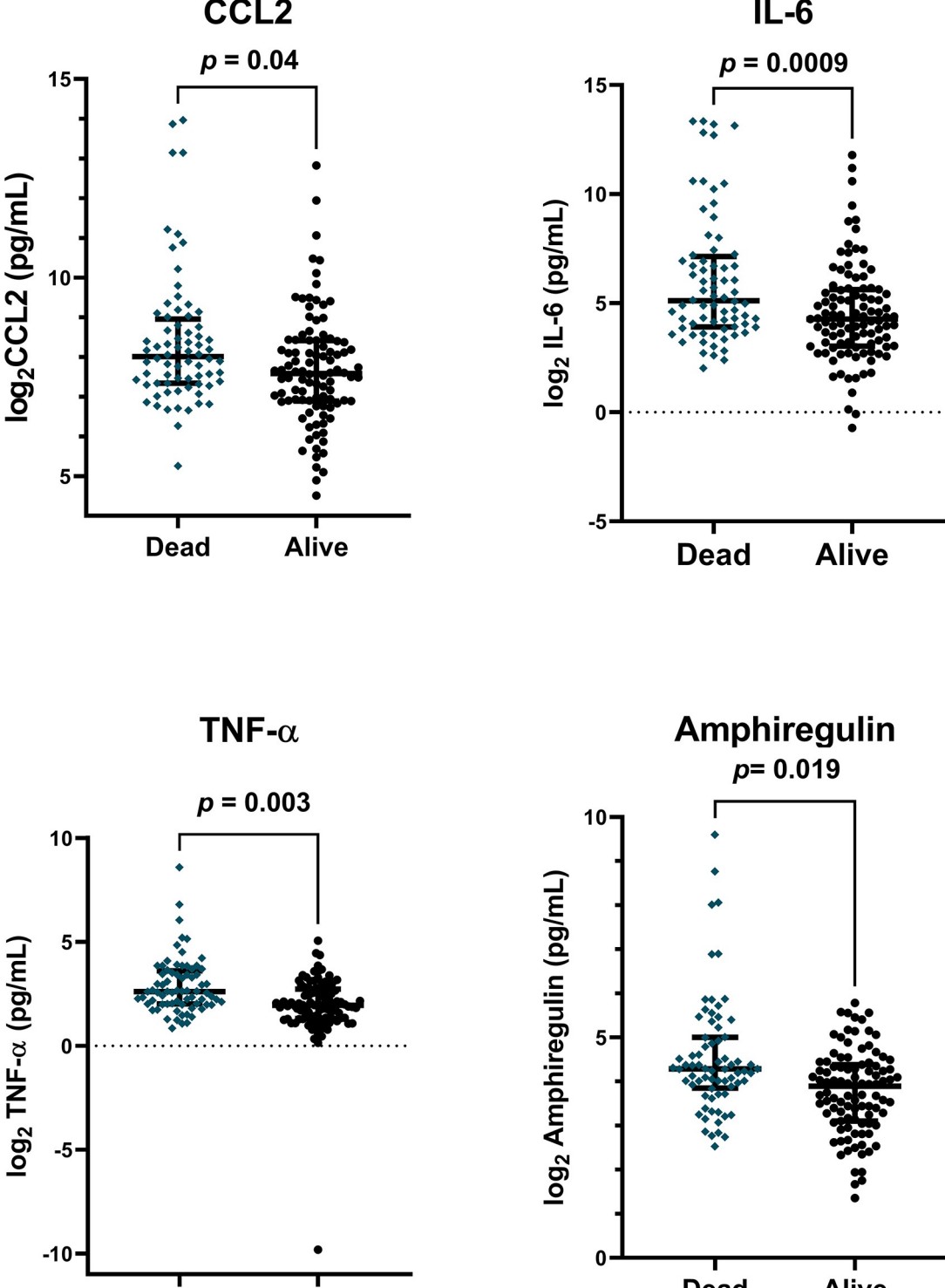

**Fig 2. Plasma proteins associated with in-hospital mortality.** Plasma protein levels were $\log_2$-transformed and adjusted for age, sex, treatment with steroids, and enrollment APACHE III score. Plasma was collected within 24h of enrollment. *p*-values reflect association of adjusted protein concentrations with in-hospital mortality using logistic regression.

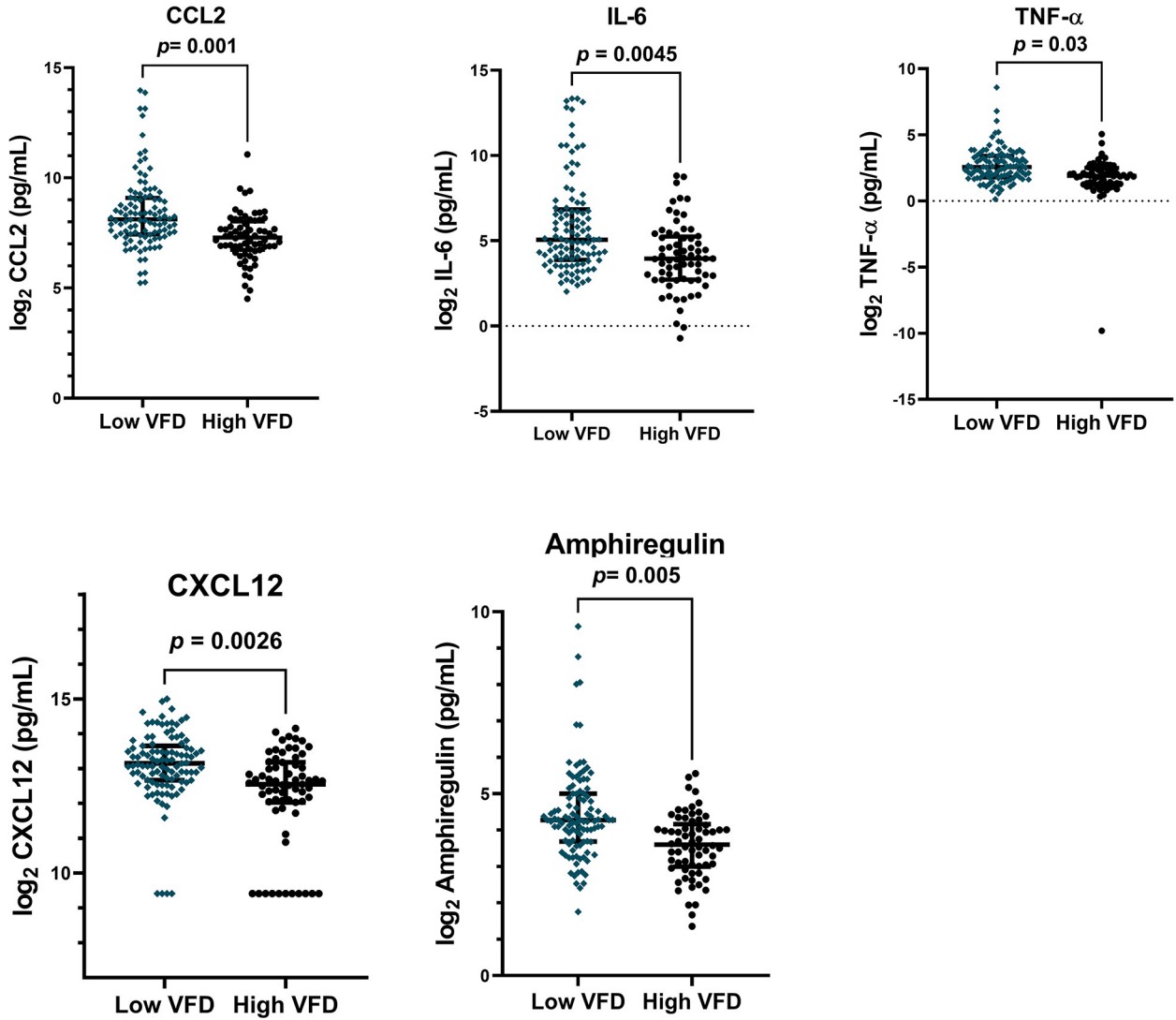

**Fig 3. Plasma proteins associated with high/low ventilator free days (VFD).** Patients were dichotomized into high (VFD≥14) or low (VFD<14) groups. Plasma protein levels were $log_2$-transformed and adjusted for age, sex, treatment with steroids, and enrollment APACHE III score. Plasma was collected within 24h of enrollment. *p*-values reflect association of adjusted protein concentrations with in-hospital mortality using logistic regression.

between $log_2$-transformed plasma protein concentration and the time interval between a patient's initial COVID+ PCR and their first sample collection (within 24h of ICU admission/ study enrollment). When we looked at each individual marker, we only found that CCL13 ($r^2$ = 0.05, p = 0.003) and CXCL12 ($r^2$ = 0.04, p = 0.01) were associated with time (**S1 Fig in S1 File**), suggesting that changes in measured plasma proteins are not entirely due to differences in disease course among our cohort.

## Patients in the ICU with COVID-19 developed radiographic features of fibroproliferation during hospitalization

We then aimed to understand the association of these same proteins with the development of radiographic fibroproliferation. In a subset of patients who had a CT scan performed after day

**Table 2. Clinical characteristics of patients evaluated for fibroproliferation within cohort (*n* = 75).** *P* value reflects two-tailed z-test for proportions and Welch's two-tailed t-test for unequal variance for continuous variables.

| Characteristics | Patients with CT scans | | |
|---|---|---|---|
| | Fibrosis (*n* = 20) | No Fibrosis (*n* = 41) | *p-value* |
| Age, average (std dev), range, yr | 51 (13), 20–74 | 55 (13), 28–75 | |
| Male Sex, *n* (%) | 11 (55%) | 29 (71%) | 0.23 |
| Ethnicity (% Hispanic) | 10 (50%) | 26 (63%) | 0.32 |
| Race (% Black) | 2 (10%) | 6 (15%) | 0.62 |
| BMI, average, $kg/m^2$ | 31.6 | 33.7 | 0.37 |
| OSH Transfer, *n* (%) | 14 (70%) | 23 (56%) | 0.3 |
| ARDS at enrollment, *n* (%) | 12 (60%) | 18 (44%) | 0.24 |
| NIV/HFNC at enrollment, *n* (%) | 2 (10%) | 15 (37%) | 0.03 (*) |
| Mechanical ventilation at enrollment, *n* (%) | 17 (85%) | 24 (59%) | 0.04 (*) |
| ECLS at enrollment, *n* (%) | 4 (20%) | 3 (7%) | 0.14 |
| Received dexamethasone, *n* (%) | 18 (90%) | 35 (85%) | 0.62 |
| Received any dose of COVID vaccination, *n* (%) | 1 (5%) | 3 (7%) | 0.73 |
| Admission P:F, median (IQR) | 50 (45–85) | 65 (50–82.5) | 0.4 |
| APACHE, median (IQR) | 87.5 (60.3–103.5) | 75 (55–90.5) | 0.18 |
| RALE score, max in first 24h, median (IQR) | 24.5 (13.75–33) | 24 (14–32) | 0.64 |
| COVID$^+$ test to proximal sample date, median (IQR), *d* | 11 (2–17.5) | 5 (1–9.75) | 0.066 |
| COVID+ test to CT scan, median (IQR), *d* | 40 (23–57.25) | 18 (5–22) | 0.016 (*) |
| ICU enrollment to CT scan, median (IQR), *d* | 23.5 (17–45.75) | 11 (0–19) | 0.026 (*) |

3 of enrollment (n = 75), CTs were evaluated by two independent chest radiologists for features of fibroproliferation including traction bronchiectasis/bronchiolectasis and peripheral reticulation (**S2 Fig in S1 File**). Of these patients, 20 were found to have fibroproliferation, 5 were indeterminate, and 41 had no evidence of fibroproliferation. There were 9 scans which were reviewed and discordant between reviewers; these were not included in the analysis. The CT scan used to make this distinction was that most proximal to discharge or death, with a median of 22.5 days following a positive COVID test. Demographics including age, race and ethnicity were similar between the two groups (**Table 2**). There were more patients who were mechanically ventilated at the time of enrollment in the group with fibroproliferation, whereas patients without fibrosis were more likely to be admitted on high flow nasal cannula or with noninvasive ventilation. Patients with fibrosis tended to have higher APACHE scores but similar P:F ratios at the time of enrollment, although this was not significant. Patients with fibroproliferation had a longer time interval between their first positive SARS-CoV-2 PCR test and the first day of sampling and included a higher percentage of patients transferred from an outside hospital (70% compared to 56%), although this was not significant (**Table 3**). Patients with fibroproliferation tended to have a longer time between their COVID+ test and enrollment sample collection, and a longer time interval between COVID+ test and CT scan (**S3 Fig in S1 File**).

## Putative markers of pulmonary fibrosis are elevated in plasma later in ICU course in patients with radiographic features of fibroproliferation

We next tested for associations between our plasma proteins of inflammation and matrix remodeling and the development of fibroproliferation. Our primary hypothesis was that markers of monocyte chemotaxis and matrix remodeling would be elevated in patients with a radiographic fibroproliferative response. When measured at 24 hours, there were no significant associations between these proteins and development of fibroproliferation (**S6 Table in**

**Table 3. Outcomes of patients evaluated for fibroproliferation.**  *P* value reflects two-tailed z-test for proportions and Welch's two-tailed t-test for unequal variance for continuous variables.

| Outcome | Patients with CT scans (*n* = 75) | | |
|---|---|---|---|
| | Fibrosis (*n* = 20) | No Fibrosis (*n* = 41) | *p*-value |
| In-hospital death, *n* (%) | 8 (40%) | 16 (40%) | 0.94 |
| VFDs, median (IQR), *days* | 0 (0–3.25) | 7 (0–24) | 0.007(*) |
| VFDs <14, *n* (%) | 18 (90%) | 27 (66%) | 0.04 (*) |
| Treatment for refractory hypoxemia | 15 (75%) | 26 (63%) | 0.37 |
| Lowest S:F in 7 days, median (IQR) | 198.8 (114.4–223.8) | 146.7 (94–200.7) | 0.6 |
| Lowest P:F in 7 days, median (IQR) | 71 (60.25–126) | 85 (72.5–95.5) | 0.8 |

VFDs- Ventilator-free days, S:F—Ratio of the $SpO_2$ to the delivered $FiO_2$

**S1 File**). However, in plasma collected 96-128h after enrollment, higher concentrations of Amphiregulin (OR 2.15, [CI 1.09–4.83], *p* = 0.038) adjusted for age, sex, APACHE score, and steroids was associated with increased odds of development of fibroproliferation (**Fig 4a**, **S7 Table in S1 File**). When we evaluated whether the change over time in plasma protein concentration between the two timepoints, we found that an increasing concentration of MMP-7 over time was associated with fibroproliferation (OR 2.14, [CI 1.12–4.46], *p* = 0.028) (**Fig 4B** and **S8 Table in S1 File**).

## Endotracheal aspirate measurements are not associated with mortality, VFDs, or fibroproliferation and are not correlated with plasma measurements

A subset of mechanically ventilated had endotracheal aspirate measurements (ETA) collected within 24h of ICU admission and again approximately 72 hours later. Again, we measured the same proteins (**S1 Table in S1 File**) via immunoassay in the ETAs. We did not find any association between any individual protein measurement and any of our specified outcomes of mortality, VFDs, or development of fibroproliferation.

We performed a correlation analysis between plasma and endotracheal aspirate measurements among patients who had paired samples at 24h (*n* = 30) **(S4 Fig in S1 File).** We found that in general, plasma and endotracheal aspirate measurements of individual proteins were not positively correlated.

## Discussion

In this cohort study of patients admitted to the ICU with severe COVID-19 pneumonia, we identified that plasma proteins previously associated with pulmonary fibrosis are elevated in patients with poor ICU outcomes including in-hospital mortality and low VFDs. A distinguishing feature of our study is the systematic review of chest imaging during hospitalization and the measurement of protein mediators in plasma at multiple timepoints. We hypothesized that proteins previously shown to be elevated in patients with pulmonary fibrosis (Amphiregulin [32, 33], MMP-9 [34, 35], MMP-7 [35], P-selectin [36], S100A12 [35], and CXCL12 [37]) would be elevated in patients who develop fibroproliferative ARDS.

We evaluated patients for fibroproliferative ARDS and identified that approximately 26% of the patients evaluated with CT scans during hospitalization had evidence of fibroproliferation, which is similar to previously described rates of post-ARDS fibrosis [38–40]. The patients who developed evidence of fibroproliferation were more likely to require mechanical ventilation at

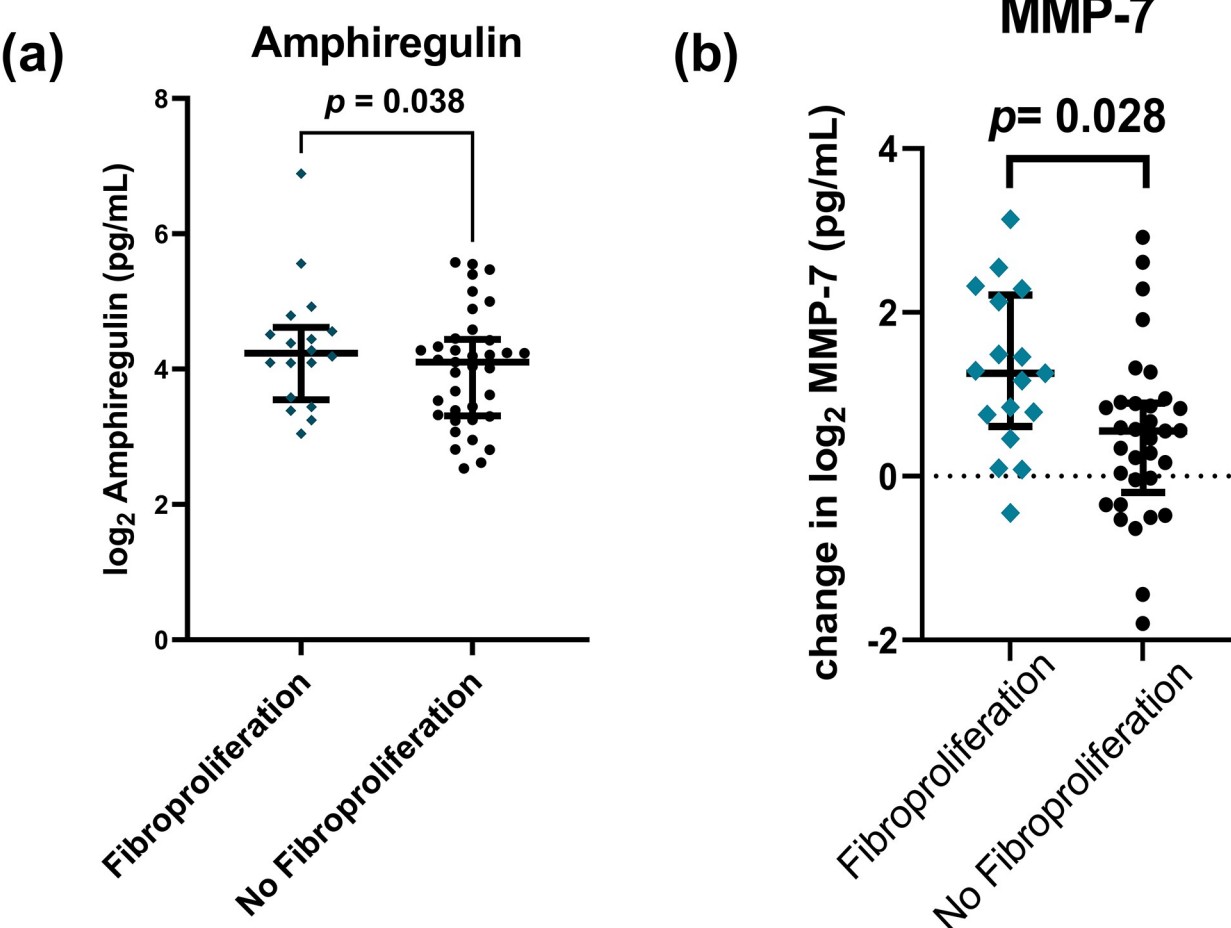

**Fig 4. Plasma proteins associated with fibroproliferation on CT scan.** a) Increased levels of amphiregulin are associated with fibroproliferation on CT scan. Plasma protein levels were $\log_2$-transformed and adjusted for age, sex, treatment with steroids, and enrollment APACHE III score. Plasma was collected 96-168h after enrollment. *p*-values reflect association of adjusted protein concentration with fibroproliferation using logistic regression. B) An increase in MMP-7 concentration over time (between plasma collected at 24h and at 96-128h) is associated with fibroproliferation. Plasma protein level was $\log_2$-transformed. *p*-value reflects association of transformed protein concentration adjusted for age, sex, treatment with steroids, and enrollment APACHE III score with fibroproliferation seen on CT scan.

the time of enrollment and had fewer VFDs, which is consistent with documented risk factors and outcomes in other ICU cohorts [24, 38–40]. We found that these patients also had elevated levels of EGFR ligand amphiregulin in their plasma.

While it has been previously documented that TNF-α and IFN-γ are associated with mortality during COVID-19 [41–44], a novel finding from our study is the association of amphiregulin with VFDs and development of fibroproliferation. Our findings support a prior study that identified amphiregulin as a cytokine associated with disease severity in COVID-19 in a proteomics screen [45]. Amphiregulin most likely has both pro-inflammatory and reparative functions. A separate study reported lower serum levels of amphiregulin in patients with severe COVID-19 [46]. The authors showed that amphiregulin may play a role in T-cell mediated tissue repair via Notch4 signaling. Amphiregulin has also been implicated in fibrosis pathogenesis in multiple tissues including the lungs [47–50]. It is secreted by T-lymphocytes and CD11c + dendritic cells with primary activity on epithelial cells [47, 50]. Epithelial cells stimulated by amphiregulin stimulate fibroblasts, and amphiregulin can also act directly on fibroblasts to produce extracellular matrix [47, 51]. Although we describe amphiregulin in a small cohort of

patients with severe COVID-19, our findings add to the growing body of evidence that amphiregulin is important in both appropriate and aberrant tissue repair mechanisms in viral ARDS.

Further work is necessary to validate these findings in the context of other clinical patient cohorts. Many published ICU COVID-19 cohorts similar to ours do not have in-depth radiographic reports, which limits the ability to validate our findings [52–56]. These markers should also be measured in patients convalescing from COVID-19, in particular those who have prolonged respiratory symptoms to understand how the levels of these proteins change in the plasma over time. More recently, cohort studies are being published describing the effects of COVID-19 on longer term outcomes, including prolonged respiratory symptoms and post-COVID-19 fibrosis [57–60].

There are several limitations to our study. Our study was performed within one geographic region, but patients were enrolled in three area hospitals with different patient demographics and ICU practice patterns which can improve the generalizability of these findings. Our cohort was recruited over a time period where clinical practice patterns regarding the use of immunosuppression varied, and plasma and ETA proteins could have been affected by treatment with dexamethasone, tocilizumab, remdesivir, or convalescent plasma. We have outlined the treatment of patients in our cohort with these in **S9 Table in S1 File**. Only 75.4% of our cohort received treatment with dexamethasone. We aimed to address this source of variability by including steroid treatment as a covariate in our adjusted model. In addition, co-infection with another organism could alter the cytokine profile of patients over time. We do not have complete records of which patients had documented ICU infections, but 99/195 were on IV antibiotics at the time of enrollment (and sample collection at the first timepoint). This does not necessarily indicate presence of a secondary infection, only clinical suspicion. We expect that the fraction of patients with a true secondary infection would be much lower. 82/195 patients had culture data analyzed. Of these, 27/82 (33%) had positive cultures from blood, stool, throat swab, pleural fluid, or endotracheal aspirates within 1 week of enrollment.

When we evaluated patients for fibroproliferative response, we identified a significant difference in the interval of time between SARS-CoV-2 test positivity and the CT scan used to classify the patient into fibrosis/no fibrosis (**S3B Fig in S1 File**). The most proximal CT scan to discharge/death was used to make this classification, and so this time interval reflects that patients with fibroproliferation had longer hospital stays. There was a nonsignificant increase in the length of time between COVID+ test to study enrollment among patients with fibroproliferation as well. A higher number of these patients were transferred from other hospitals to our institution for specialized ARDS care, although this was not statistically significant (**Table 3**). It is possible that the differences we observed in plasma proteins in patients who developed fibroproliferation is due in part to a difference in time of disease course. However, when we measured the association between COVID+ test to sample collection against the concentration of proteins measured, we found an association only with MCP-4 and CXCL12. These proteins were not associated with the fibroproliferative response, suggesting that it is not only timing of disease course that leads to this association. Finally, it is unclear how many of the patients with radiographic features of fibrosis during hospitalization will go on to have persistent fibrotic changes. This is an active area of investigation in the field.

Our study defines a cohort of patients with COVID-19 who develop radiographic features of fibroproliferation during illness and compares them to patients who did not develop these features despite critical illness and severe hypoxemic respiratory failure. There may be individual immune host factors that increase the likelihood that a patient will develop fibrotic remodeling after injury. Further studies are necessary to identify the pathways that are involved. Our findings have an implication for the development of biomarkers that may predict patients at risk for developing pulmonary fibroproliferation following hypoxemic respiratory failure, as

well as elucidating potential pathways that could be targeted to improve outcomes of patients with viral induced ARDS.

## Methods

### Study design

This is a prospective cohort study nested within a larger study that has been previously described [28, 61]. We describe a cohort of patients admitted to the ICU from three hospitals in Seattle, WA between March 16, 2020 and May 16, 2021 with clinical suspicion for COVID-19. Patients were included if they tested positive for COVID-19 by PCR nasal swab and had hypoxemic respiratory failure requiring organ support as defined by need for oxygen supplementation of high flow nasal cannula, noninvasive ventilation, or invasive mechanical ventilation at the time of enrollment. Patients were excluded if they tested negative for COVID-19, had pre-existing interstitial lung disease, age $\leq$ 18, pregnancy, or current incarceration. Some patients were admitted to the ICU for monitoring but did not have hypoxemic respiratory failure, and so only patients who were on supplemental oxygen, high flow nasal cannula, non-invasive ventilation, or invasive mechanical ventilation were included in the analysis ($n$ = 206). A total of 195 patients had plasma collected for analysis and met the inclusion/exclusion criteria for the study.

All subjects were enrolled and had plasma collected within 24 hours of enrollment. A subset of patients had plasma collected 96-128h after ICU admission. Patients who were intubated had endotracheal aspirates collected within 24h of ICU admission. SARS-CoV-2 positive subjects were classified based on a positive SARS-CoV-2 RT-PCR nasal swab clinical test at the time of enrollment. Subjects did not have multiple SARS-CoV-2 RT-PCR nasal swab tests done during the study and so we cannot comment on whether they remained PCR positive at the second time-point. However, a study done in a similar cohort suggests that the majority of critically ill patients will remain PCR positive a median of 13 days after initial test positivity [62]. Subjects were enrolled under a waiver of consent which was approved and supervised by the University of Washington IRB (Human Subjects Division Study: 9763). ETAs were also collected from critically ill patients with SARS-CoV-2 supported on invasive mechanical ventilation from Virginia Mason Franciscan Health Hospital (Benaroya Research Institute IRB number: 20–036). Patients/legal representatives were consented for data usage; if consent was withdrawn, patient data and samples were removed from the study. Some authors had access to information that could identify individual participants during data collection; this data was stored in a secure online database (RedCap).

### Analysis of CT scans

A subset of patients had chest CTs done between day 3 and discharge/death. These images were ordered as part of routine clinical care and were not pre-specified. Images obtained closest to discharge or death were reviewed by a blinded chest radiologist for the presence of fibrosis. The chest CT scans were analyzed according to predominant pattern (ground glass opacities, consolidation, linear densities, reticulation, honeycombing, traction bronchiectasis, cysts, pneumatoceles), distribution (craniocaudal, axial, anterior/posterior), and overall disease extent (none, <5%, >5%). Finally, the pattern was identified as either fibrotic or non-fibrotic as determined by two independent chest radiologists using the same criteria.

### Plasma and endotracheal aspirate cytokine/chemokine measurements

Cytokines and chemokines were measured from blood collected in cell preparation tubes (CPT) containing buffered sodium citrate or EDTA and a density gradient solution for the isolation of plasma and mononuclear cells simultaneously. The tubes containing whole blood

were centrifuged at 1500g for 20 minutes, at which point the tube contained layers of plasma, mononuclear cells, and platelets, followed by granulocytes and erythrocytes separated by a gel layer. The plasma was aliquoted in 0.5 mL aliquots and frozen down, carefully avoiding the mononuclear cell layer. The aliquots were stored at -80C until further analysis. The proteins were measured using electrochemiluminescent immunoassays per the manufacturer's instructions (V-Plex Proinflammatory Panel 1 (K15049D); V-Plex Chemokine Panel 1 (K15047D); V-Plex Cytokine Panel 1 (K15050D)). Mesoscale discovery U-Plex kits were used for total MMP-9, MMP-7, SDF-1a, TGF-β1, S100A12, and P-selectin. These plasma samples were collected in a sodium citrate tube but otherwise were processed the same way as those previously described. All plasma samples underwent two freeze-thaw cycles prior to analysis. Analytes that did not meet any of the following quality control parameters were excluded from subsequent analysis: 1) intraplate % CV > 25%; 2) interplate % CV > 25%; or 3) > 10% of samples with a measurement below the lower limit of detection. For values that were below the lower limit of detection per assay (S1 Table in S1 File), concentrations were imputed by using one-half of the lower limit of detection.

ETAs were obtained by suctioning the endotracheal tube after instilling 10 mL of normal saline. The collected aspirate fluid was immediately mixed with an equal volume of 0.1% dithiothreitol and then placed on ice for 15 minutes to promote sample homogenization. Samples were then filtered through a 70 μm cell-strainer by gravity and flow-through was centrifuged at 400 x *g* for 10 minutes. The flow-through was immediately aliquoted and stored at -80°C until use. The samples underwent two freeze-thaw cycles prior to immunoassay analysis. We applied the V-Plex Pro-inflammatory (K15049D), V-Plex chemokine (K15047D), and U-Plex immunoassays on ETA samples as described above.

## Quantification and statistical analyses

### Outcome definitions

We abstracted clinical data from the electronic medical record into standardized case report forms. ARDS was defined by the 2012 Berlin definition and chest x-rays were adjudicated for ARDS by a board-certified radiologist blinded to the primary data. APACHE III score was calculated based on the original instrument [63]. VFDs were defined as the total number of days alive and free of invasive mechanical ventilation in the 28 days following ICU admission [64]. Patients who died prior to day 28 were considered to have zero VFDs. RALE score was calculated as previously described [65].

### Statistical analyses

Our primary analysis tested for association between plasma or ETA protein concentrations and in-hospital mortality, VFDs≥14 ("high") and the development of fibroproliferation based on CT scan. We used multivariable logistic regression and adjusted the $log_2$-transformed protein concentration for age, sex, treatment with dexamethasone, and enrollment APACHE III score. This regression analysis was performed in R using the glm function with the "binomial" family to specify a logistic regression where the dependent variable was binary (death, fibrosis, or high/low VFDs).

All analyses were performed in R version 4.1.1 and graphs were created in GraphPad Prism version 8.4.3.

## Supporting information

**S1 File.**
(DOCX)

## Acknowledgments

The authors thank Sharon Sahi, Carolyn Brager, Sana S. Sakr, Neall Koetje, Ashley Garay, Brian Lee, Leslie Lazar, Sonya Homami, Grigory Loginov, Jana Zahlan, Hana Morris, Jada Roth and the Benaroya Research Institute COVID-19 Research Team for sample collection and processing. The authors also thank the patients, families, surrogates, and hospital clinical staff who contributed to this work during the COVID-19 pandemic.

## Author Contributions

**Conceptualization:** Sarah E. Holton, Sudhakar Pipavath, Mark M. Wurfel, Carmen Mikacenic.

**Data curation:** Sarah E. Holton, Mallorie Mitchem, Eric D. Morrell, Cate Speake, Uma Malhotra, Carmen Mikacenic.

**Formal analysis:** Sarah E. Holton, Hamid Chalian, Sudhakar Pipavath, Eric D. Morrell, Carmen Mikacenic.

**Funding acquisition:** Jessica A. Hamerman, Steven E. Ziegler.

**Investigation:** Sarah E. Holton, Mallorie Mitchem, Sudhakar Pipavath, Eric D. Morrell, Mark M. Wurfel, Carmen Mikacenic.

**Methodology:** Sarah E. Holton, Sudhakar Pipavath, Eric D. Morrell, Pavan K. Bhatraju, Mark M. Wurfel.

**Project administration:** Cate Speake, Steven E. Ziegler.

**Resources:** Carmen Mikacenic.

**Supervision:** Eric D. Morrell, Steven E. Ziegler.

**Validation:** Sarah E. Holton.

**Visualization:** Sarah E. Holton.

**Writing – original draft:** Sarah E. Holton, Carmen Mikacenic.

**Writing – review & editing:** Sarah E. Holton, Mallorie Mitchem, Hamid Chalian, Sudhakar Pipavath, Eric D. Morrell, Pavan K. Bhatraju, Jessica A. Hamerman, Cate Speake, Uma Malhotra, Mark M. Wurfel, Steven E. Ziegler, Carmen Mikacenic.

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
