## [Decision Letter · Decision Letter 0]

22 Jun 2023

PONE-D-23-11100

Mediators of monocyte chemotaxis and matrix remodeling are associated with the development of fibrosis in patients with COVID-19

PLOS ONE

Dear Dr. Holton,

Thank you for submitting your manuscript to PLOS ONE. After careful consideration, we have decided that your manuscript does not meet our criteria for publication and must therefore be rejected. This is based on the reviewers' concerns regarding the timing of the sample acquisition and imaging, the high rate of plain film chest radiographs and others. Please refer to the suggestions by the reviewers for detailed information.

This is an "OPEN REJECTON". For major revisions PLOS offeres a time frame of 45 days which might be a short time for the additional analyses requested by by the reviewers. If you think that a reanalysis of your data according with the suggestions of the reviewers is possible, a resubmission of the revised manuscript is welcome. I am sorry that we cannot be more positive on this occasion, but hope that you appreciate the reasons for this decision.

Kind regards,

Gernot Zissel, Ph.D.

Academic Editor

PLOS ONE

Reviewers' comments:

Reviewer's Responses to Questions

**Comments to the Author**

1. Is the manuscript technically sound, and do the data support the conclusions?

Reviewer #1: Yes

Reviewer #2: No

2. Has the statistical analysis been performed appropriately and rigorously? 

Reviewer #1: Yes

Reviewer #2: No

3. Have the authors made all data underlying the findings in their manuscript fully available?

Reviewer #1: Yes

Reviewer #2: Yes

4. Is the manuscript presented in an intelligible fashion and written in standard English?

Reviewer #1: Yes

Reviewer #2: Yes

5. Review Comments to the Author

Reviewer #1: Thank you very much for giving me the opportunity to review this manuscript.

In this work, the authors monitored chemokine and fibrotic markers in the blood of patients that developed, or not, early fibrotic features on the CT scan within 24 h after ICU admission.

While ,the statistical methodology seems robust several points dampened my enthusiasm.

1)Description of the population:

It would be important for the reader to get more information regarding the treatment or the vaccination status of the patients. These factors could definitely affect the cytokines quantification

Did the patients continue to receive dexamethasone during plasma cytokine quantification

How many patients received other immunosuppressive drugs etc.….

I was surprised to see that only 50 and 75 % of the patient’s received dexamethasone. Could the authors explain this result.

2) Could the authors detail the infection status of these patients?

-COVID status (PCR negative or positive)

- Other ICU acquired infection

3) I’m not complete sure that the section about the trachea aspiration is really useful considering all the limitation related to this kind of measurement.

MMP7, MMP9, CCL2 appear to be associated with fibrotic features onset.

Did the authors run any analysis to assess the relationship between the cytokines course or the and some outcomes like mortality or time to extubating etc……

Reviewer #2: In the manuscript entitled “Mediators of monocyte chemotaxis and matrix remodelling are associated with the development of fibrosis in patients with COVID-19” Holton and colleagues analysed the plasma and endotracheal aspirates of COVID-19 patients admitted to ICU. Using chest imaging they evaluated each patient for the presence of lung fibrotic features, and subsequently measured markers of matrix remodelling and monocyte migration. They identified CCL-2/MCP-1, CCL13/MCP-4, Amphiregulin, MMP-7 and MMP-9 increased in the plasma of patients with evidence of fibrosis and CCL2/MCP-1 in the endotracheal aspirates. The authors conclude these are relevant biomarkers to tissue remodelling and monocyte recruitment occurring early during the development of lung fibrosis induced by COVID-19.

This work is an important topic to address in the field. However, there are several flaws with the paper that need a large effort to address. Of particular concern is the difference in the timing between testing, sampling and imaging the study groups, the reliance on non-CT imaging for fibrotic diagnosis, and the presenting of this data as early, predictive markers of fibrosis. This makes it difficult to be sure the conclusions of the study are supported by the data presented. From this prospective, I cannot recommend the paper for publication in its current form. I would encourage the authors to reanalyse their data, taking account of the concerns listed below.

Major concerns

1. The timing of the sample acquisition and imaging is an issue.

First, the time between a positive COVID-19 test and sampling is significantly longer in the non-survivor group (Supp Fig 2a). Why is the data in Supp Fig 2 stratified based on survivability? It does not appear the data in Figs 3 and 4 are segregated in this way, so why are they stratified here? Could the data in Supp Fig 2 be graphed with both survivor and non-survivor data combined, to see whether COVID+ test to enrolment (sampling) is significantly different overall?

In table 1, timing between positive test and sample date is median 5 days vs 12.5 days for non-fibrotic vs fibrotic. This makes it difficult to exclude the possibility that changes in plasma proteins are not a fibrotic difference but rather due to the fibrotic group being later in COVID-19 disease course. This is a concern for the study conclusion, that these proteins are early markers of COVID-19 induced lung fibrosis. Fibroproliferation happens at exactly the time-point seen in table 1 (days 10-21), with a higher percentage of mechanically ventilated (intubated) patients here. The protein changes are probably more so markers of where the patient is in the ARDS pathogenesis disease time-course- a CT scan performed at the time of sampling could also likely give you this information. Although the authors seek to address this through the explanation of hospital transfers, this does not adequately address the concerns raised.

Secondly, the difference between a positive test date and date of imaging is significantly longer in the fibrotic groups (Supp Fig 2). Patients may have had a CT scan as close as 3 days to the initial sample and no patient had any imaging after discharge. i.e. the imaging is very close to time of sampling, and there has been no evidence of “post-ARDS fibrosis”, which would be the clinical end point that matters most for long-term survivors. This should be measured several weeks into convalescent to give time for matrix remodelling and lung repair to have occurred. i.e. it needs to be established fibrosis. There are no firm guidelines but in ILD centres we are generally assessing for post-COVID fibrosis 6 months after infection. We tend to assess for post-ARDS fibrosis at least 6 weeks after discharge and then again at 6 months to determine whether this is established fibrosis- COVID-19 and ARDS fibrosis can improve surprising amounts.

2. The use of plain film chest radiographs for the determination of fibrosis is a problem, as it has extremely poor sensitivity-a CT scan is the gold standard diagnosis tool and only 32/119 patients had CT scans. While there are many good reasons CT scans cannot be performed, the authors should ensure that the decision to include non-CT imaging does not confound the study conclusion.

Despite the statement in lines 133-136, examination of Supp Table 3 demonstrates patients diagnosed without CT scans did not show the same changes as CT scans alone, (e.g. CCL13 p=0.04 with CT scan compared to p=0.62 for without CT scans). For CCL2, the results, while significant overall, are not significant for CT (p=0.1) or without CT (p=0.95)- I am unsure how this could have occurred? Further, the bias towards diagnosis of fibrosis in CT vs non-CT described in the discussion (lines 238-242) advocates that non-CT analysed imaging should be dropped from the study altogether. I suggest including only CT scan patient data in the results to be sure of the diagnosis and hence the conclusion of the study.

3. Primarily the authors discuss their findings as early biomarkers of fibrosis that are present soon after ICU admission (e.g. Lines 21-23 of abstract and line 201-202 and 261-263 of discussion). However, as samples were taken many days after a COVID+ test (see point 1) it is likely that many of these patients could have ARDS fibroproliferation already at ICU admission (which is why patients have deteriorated to needing ICU admission). This is a pathological process within lots of patients who develop worsening ARDS and can start from 7 days and continue up to 3 weeks after injury. Therefore, I would argue that the biomarker measurements are not predictive biomarkers of fibrosis but markers of severity of ARDS fibroproliferation at that time point. We already know that worse ARDS= worse outcome, and there are ways of measuring ARDS severity clinically, which table 1 shows are higher in the fibrosis group at time of sampling (Berlin, P/F ratio, APACHE II score- as a side note, can statistical tests be performed here to determine whether any are statistically significant?).

To demonstrate any prognostic utility the authors would need to show that these biomarkers identify mortality or ventilator free days better than clinical tools criteria. Ideally sampling should be done on hospital admission to give opportunity to therapeutically immunomodulate. Failing that, the authors should reword the text throughout to make clear these results are not predictive of fibrosis but markers of severity.

Can the authors provide supplementary data/sub analysis from the patients who had a CT scan within 72 hours of ICU admission and how many had elevated biomarkers and radiographic fibrosis at this timepoint (i.e. were sampled in the first few days)? This would be important to explore whether they are just diagnostic markers of fibrosis (of which there are several already known).

4. The authors make their fibrotic diagnosis using a single, blinded specialist thoracic radiologist. Given this is the primary outcome of the study, you would usually want two people to independently arrive at the same conclusion, as these are holistic judgement interpretations essentially based upon a variety of radiological features.

In addition, 17 patients did not have their images reviewed, just the original radiologist report provided (therefore the decision was not completely blinded). I would suggest these samples need removing from the analysis.

Other comments

-The authors assert that these proteins are monocyte/macrophage derived. While a very plausible theory, no data has been shown on these cell types directly in this study. Can these factors expression be shown from PBMC isolations from patients? It would be understandable that these samples don’t exist and are not acquirable, in which case the text should be altered to address this.

-Lines 114-117- when looking at supp table 2 normalising for steroids (right hand column) there is no significant difference for IL-6 and TNFa as described. In fact, in contrast to the text, there is a difference for CCL-2, CCl13, Amphiregulin, MMP9 and MMP7. The data points used for IL-6 and TNFa in this case appear to come from the unadjusted data.

-Lines 74-75 and Table 1- Can you perform statistical tests here to demonstrate whether the characteristics, particularly APACHE III, P:F ratios, and time to COVID+ test proximal sample date, are statistically significant between groups?

-The referencing of monocyte and macrophage evidence in COVID-19, fibrosis and ARDS could be improved. There are a number of strong studies on this that could be included.

-Supp Table 5 and Supp Table 6 is misquoted in the text (lines 145-146 and 148-149). Lines 155-157 should quote Supp Table 6 instead of Supp Table 4.

6. PLOS authors have the option to publish the peer review history of their article (what does this mean?). If published, this will include your full peer review and any attached files.

Reviewer #1: No

Reviewer #2: No

- - - - -

---

## [Author Response · Author response to Decision Letter 0]

19 Jan 2024

Reviewer 1 General Comments: Thank you very much for giving me the opportunity to review this manuscript.

In this work, the authors monitored chemokine and fibrotic markers in the blood of patients that developed, or not, early fibrotic features on the CT scan within 24 h after ICU admission.

While ,the statistical methodology seems robust several points dampened my enthusiasm.

Response to Reviewer 1: Thank you for taking the time to extensively review our manuscript and for providing your thoughtful comments. We address each of your specific comments below.

Reviewer Comment #1: Description of the population:

It would be important for the reader to get more information regarding the treatment or the vaccination status of the patients. These factors could definitely affect the cytokines quantification. Did the patients continue to receive dexamethasone during plasma cytokine quantification

How many patients received other immunosuppressive drugs etc.….

I was surprised to see that only 50 and 75 % of the patient’s received dexamethasone. Could the authors explain this result.

Response to Comment #1: Thank you for bringing up this important point regarding the vaccination status of the patients. Only 7 patients had received one dose of any COVID vaccination at the time of sample collection and we have clarified this in the text (Lines 71-73) as well as Table 1. This very low number reflects two characteristics of our cohort. First, the cohort was enrolled between March 2020 and May 2021 which is prior to wide release of the COVID vaccination in December 2020. Second, many of our patients came from vaccine-hesitant populations in regions of Washington, Wyoming, Montana, Idaho, and Alaska. 

The administration of dexamethasone or other immunosuppressive medication can certainly affect cytokine production as you have noted. The majority of patients who did not receive dexamethasone in our cohort were enrolled prior to the publication of the RECOVERY trial in February 2021. We have extracted other immunosuppressive treatments that could affect cytokine production from patients records including tocilizumab, methylprednisolone, prednisone, hydroxychloroquine, and remdesivir for each and the dosing relative to timing of sample collection in our supplementary table 9. 

Reviewer Comment #2: Could the authors detail the infection status of these patients?

-COVID status (PCR negative or positive)

- Other ICU acquired infection

Response to Comment #2: Thank you for bringing up this important point. All of the patients contained within our cohort were COVID PCR positive at the time of sample collection at the first timepoint and this has been clarified in the methods section. The patients were not tested repeatedly while in the ICU due to limited capacity of clinical testing during the height of the pandemic so we do not know the persistence of SARS-CoV2 PCR positivity of the patients at the second timepoint. We suspect that the patients were still PCR positive at the second timepoint (96-128h after enrollment) given their severe critical illness. We have noted this in the methods section (lines 258-263) and cited an article that reports 74.1% of patients (n=108) remaining RT-PCR positive a median of 13 days after initial positive PCR test in a similar critically-ill cohort (Funk DJ et al Persistence of live virus in critically ill patients infected with SARS-CoV-2: a prospective observational study, Crit Care 2022 26(1):10).

We do not have complete records of which patients had documented ICU infections for the entire cohort. However, 99/195 were on IV antibiotics at the time of enrollment (sample collection V1). This does not necessarily indicate presence of a secondary infection, only clinical suspicion. In 82 patients who had culture data analyzed, only 27 had positive cultures from blood, stool, throat swab, pleural fluid, or endotracheal aspirates. This rate of 33% when projected over our entire cohort would suggest only 66 patients had a secondary infection at the time of sampling. We have similarly included this in the discussion (lines 207-214).

Reviewer Comment #3: I’m not complete sure that the section about the trachea aspiration is really useful considering all the limitation related to this kind of measurement.

MMP7, MMP9, CCL2 appear to be associated with fibrotic features onset.

Did the authors run any analysis to assess the relationship between the cytokines course or the and some outcomes like mortality or time to extubating etc……

Response to Comment #3: Thank you again for this excellent point. We included the endotracheal aspirate data in part to show the discordance between plasma and endotracheal aspirate measurements. We also wanted to include these negative results as there is significant debate in the field as to the value of these measurements (Mikacenic et al 2023 Reply: Research Bronchoscopy Standards and the need for Non-invasive sampling of the Failing Lungs, Annals ATS PMID 37776284). Also, due to limitations on personnel and PPE at our hospitals during the COVID pandemic, bronchoalveolar lavage samples were not able to be collected on patients with COVID-19. We have removed the endotracheal aspirate data and incorporated this into our discussion section (lines 152-160). We have included the correlation analysis between plasma and ETA measurements in Supplemental Figure 4. 

We have reframed our manuscript around clinical endpoints associated with prolonged respiratory failure and mortality in addition to radiographic features of fibroproliferation. We have assessed the relationship between cytokine measurements in the plasma and in-hospital mortality and high/low ventilator-free days. These analyses are shown in Figures 2 and 3.

Reviewer 2 General Comments: In the manuscript entitled “Mediators of monocyte chemotaxis and matrix remodelling are associated with the development of fibrosis in patients with COVID-19” Holton and colleagues analysed the plasma and endotracheal aspirates of COVID-19 patients admitted to ICU. Using chest imaging they evaluated each patient for the presence of lung fibrotic features, and subsequently measured markers of matrix remodelling and monocyte migration. They identified CCL-2/MCP-1, CCL13/MCP-4, Amphiregulin, MMP-7 and MMP-9 increased in the plasma of patients with evidence of fibrosis and CCL2/MCP-1 in the endotracheal aspirates. The authors conclude these are relevant biomarkers to tissue remodelling and monocyte recruitment occurring early during the development of lung fibrosis induced by COVID-19.

This work is an important topic to address in the field. However, there are several flaws with the paper that need a large effort to address. Of particular concern is the difference in the timing between testing, sampling and imaging the study groups, the reliance on non-CT imaging for fibrotic diagnosis, and the presenting of this data as early, predictive markers of fibrosis. This makes it difficult to be sure the conclusions of the study are supported by the data presented. From this prospective, I cannot recommend the paper for publication in its current form. I would encourage the authors to reanalyse their data, taking account of the concerns listed below.

Response to Reviewer 2:

Thank you for your thorough review and comments on our manuscript. We have addressed your comments and made extensive changes both in the analysis of our data and the framing of our conclusions. We now discuss radiographic fibroproliferation or fibroproliferative response rather than “fibrosis” to help clarify that we are not making generalizations regarding the development of post-COVID fibrosis. The major concern of the timing between testing, sampling, and imaging of the study groups is one that is difficult to overcome given the retrospective nature of the study. All CT scans were clinically indicated rather than predetermined, and physician practice varied during different times of the pandemic. Nevertheless, there is scant literature in the field as to biomarkers associated with fibroproliferation in ARDS and we believe that despite these limitations this is an important contribution. In addition to further address these concerns, we have performed considerable new analyses and rewritten the majority of the manuscript to focus on other clinical outcomes that reflect the development of pulmonary remodeling/fibrosis such as in-hospital mortality and prolonged ventilator-free days. We have specifically addressed your major and minor concerns individually below.

Reviewer 2 Major Comment 1: The timing of the sample acquisition and imaging is an issue.

First, the time between a positive COVID-19 test and sampling is significantly longer in the non-survivor group (Supp Fig 2a). Why is the data in Supp Fig 2 stratified based on survivability? It does not appear the data in Figs 3 and 4 are segregated in this way, so why are they stratified here? Could the data in Supp Fig 2 be graphed with both survivor and non-survivor data combined, to see whether COVID+ test to enrolment (sampling) is significantly different overall?

In table 1, timing between positive test and sample date is median 5 days vs 12.5 days for non-fibrotic vs fibrotic. This makes it difficult to exclude the possibility that changes in plasma proteins are not a fibrotic difference but rather due to the fibrotic group being later in COVID-19 disease course. This is a concern for the study conclusion, that these proteins are early markers of COVID-19 induced lung fibrosis. Fibroproliferation happens at exactly the time-point seen in table 1 (days 10-21), with a higher percentage of mechanically ventilated (intubated) patients here. The protein changes are probably more so markers of where the patient is in the ARDS pathogenesis disease time-course- a CT scan performed at the time of sampling could also likely give you this information. Although the authors seek to address this through the explanation of hospital transfers, this does not adequately address the concerns raised.

Secondly, the difference between a positive test date and date of imaging is significantly longer in the fibrotic groups (Supp Fig 2). Patients may have had a CT scan as close as 3 days to the initial sample and no patient had any imaging after discharge. i.e. the imaging is very close to time of sampling, and there has been no evidence of “post-ARDS fibrosis”, which would be the clinical end point that matters most for long-term survivors. This should be measured several weeks into convalescent to give time for matrix remodelling and lung repair to have occurred. i.e. it needs to be established fibrosis. There are no firm guidelines but in ILD centres we are generally assessing for post-COVID fibrosis 6 months after infection. We tend to assess for post-ARDS fibrosis at least 6 weeks after discharge and then again at 6 months to determine whether this is established fibrosis- COVID-19 and ARDS fibrosis can improve surprising amounts.

Response to Major Comment 1:

We agree that the timing discrepancies between sample collection and imaging is the biggest issue with our dataset. As such, we have reframed our manuscript such that our primary outcomes are not dependent on radiographic definitions. 

We have re-graphed the data in Supplemental Figure 3 so that the data are not stratified based on survivability. 

It is very possible that changes in plasma proteins are due to differences in the COVID-19 disease course and are reflective of a different phase of ARDS. We have adjusted our discussion of this in the text. We have reviewed all CT scans performed during hospitalization in the cohort and found that at the time of sample collection, no patient had evidence of a fibrotic pattern on their CT scan (this is evaluated now by two independent chest radiologists). This makes it more likely that the patients who develop fibrosis are different in some way but we can only speculate rather than make a conclusion based on our study design and results. 

We used the term “fibrosis” too loosely in our original manuscript. We did not have the ability to follow patients or make sample collections after discharge from the hospital which prevented us from understanding who had true post-ARDS/post-COVID fibrosis. We used a radiographic definition during hospitalization which is more reflective of the fibroproliferative phase of ARDS. We have adjusted our definitions in the manuscript and highlighted this important discrepancy in the discussion. A significant number of patients within our cohort died from overwhelming ARDS/respiratory failure and other associated complications. We think that a strength of our manuscript is that these patients are included whereas in many studies looking at post-ARDS fibrosis they would be excluded. We feel it is important to study this group of patients to determine what modifiable factors may lead to death from fibrotic ARDS. 

To address whether the proteins measured reflect where the patients are in the disease course, we have plotted the log2-transformed plasma concentrations of each marker against the interval between the patient’s initial covid+ PCR and the V1/enrollment sample collection. Then, simple linear regression was used to determine if there was any linear association. We found that only MCP-4 (p=0.003) and CXCL12 (p=0.01) had a linear association with time. This suggests that the protein mediators discussed in this article (with the exception of MCP-4 and CXCL12) are not entirely dependent on disease course. We have discussed this in the text (Lines 102-110) and included it in Supplemental Figure 1. To be as thorough as possible, the date of COVID+ test was extracted from the patient’s chart and included results collected outside of our system including at primary care clinics and urgent care. 

Reviewer 2 Major Comment 2: The use of plain film chest radiographs for the determination of fibrosis is a problem, as it has extremely poor sensitivity-a CT scan is the gold standard diagnosis tool and only 32/119 patients had CT scans. While there are many good reasons CT scans cannot be performed, the authors should ensure that the decision to include non-CT imaging does not confound the study conclusion.

Despite the statement in lines 133-136, examination of Supp Table 3 demonstrates patients diagnosed without CT scans did not show the same changes as CT scans alone, (e.g. CCL13 p=0.04 with CT scan compared to p=0.62 for without CT scans). For CCL2, the results, while significant overall, are not significant for CT (p=0.1) or without CT (p=0.95)- I am unsure how this could have occurred? Further, the bias towards diagnosis of fibrosis in CT vs non-CT described in the discussion (lines 238-242) advocates that non-CT analysed imaging should be dropped from the study altogether. I suggest including only CT scan patient data in the results to be sure of the diagnosis and hence the conclusion of the study.

Response to Major Comment 2: We agree that the use of plain film chest radiographs is not the gold standard for assessing fibrosis. As such, we have excluded patients who did not have CT scans for our fibrosis outcome and redone our graphs, calculations, and statistical tests. 

Reviewer 2 Major Comment 3: Primarily the authors discuss their findings as early biomarkers of fibrosis that are present soon after ICU admission (e.g. Lines 21-23 of abstract and line 201-202 and 261-263 of discussion). However, as samples were taken many days after a COVID+ test (see point 1) it is likely that many of these patients could have ARDS fibroproliferation already at ICU admission (which is why patients have deteriorated to needing ICU admission). This is a pathological process within lots of patients who develop worsening ARDS and can start from 7 days and continue up to 3 weeks after injury. Therefore, I would argue that the biomarker measurements are not predictive biomarkers of fibrosis but markers of severity of ARDS fibroproliferation at that time point. We already know that worse ARDS= worse outcome, and there are ways of measuring ARDS severity clinically, which table 1 shows are higher in the fibrosis group at time of sampling (Berlin, P/F ratio, APACHE II score- as a side note, can statistical tests be performed here t

---

## [Decision Letter · Decision Letter 1]

11 Apr 2024

PONE-D-23-11100R1Mediators of monocyte chemotaxis and matrix remodeling are associated with mortality and pulmonary fibroproliferation in patients with severe COVID-19PLOS ONE

Dear Dr. Holton,

Thank you for submitting your manuscript to PLOS ONE. After careful consideration, we feel that it has merit but does not fully meet PLOS ONE’s publication criteria as it currently stands. Therefore, we invite you to submit a revised version of the manuscript that addresses the points raised during the review process.

One reviewer asks for additional information regarding statistics and methods. Please amend your manuscript according to the queries of the reviewer.

We look forward to receiving your revised manuscript.

Kind regards,

Gernot Zissel, Ph.D.

Academic Editor

PLOS ONE

Journal Requirements:

1. Please ensure that your manuscript meets PLOS ONE's style

requirements, including those for file naming. The PLOS ONE style templates can

be found at

and

2. Please note that PLOS ONE has specific guidelines on code

sharing for submissions in which author-generated code underpins the findings

in the manuscript. In these cases, all author-generated code must be made

available without restrictions upon publication of the work. Please review our

guidelines at

https://journals.plos.org/plosone/s/materials-and-software-sharing#loc-sharing-code

and ensure that your code is shared in a way that follows best practice and

facilitates reproducibility and reuse.

3. Please be informed that funding information should not

appear in the Acknowledgments section or other areas of your manuscript. We

will only publish funding information present in the Funding Statement section

of the online submission form. Please remove any funding-related text from the

manuscript.

4. Your ethics statement should only appear in the Methods

section of your manuscript. If your ethics statement is written in any section

besides the Methods, please move it to the Methods section and delete it from

any other section. Please ensure that your ethics statement is included in your

manuscript, as the ethics statement entered into the online submission form

will not be published alongside your manuscript.

5. Please include captions for your Supporting Information

files at the end of your manuscript, and update any in-text citations to match

accordingly. Please see our Supporting Information guidelines for more

information: http://journals.plos.org/plosone/s/supporting-information.

Additional Editor Comments (if provided):

Reviewers' comments:

Reviewer's Responses to Questions

**Comments to the Author**

1. If the authors have adequately addressed your comments raised in a previous round of review and you feel that this manuscript is now acceptable for publication, you may indicate that here to bypass the “Comments to the Author” section, enter your conflict of interest statement in the “Confidential to Editor” section, and submit your "Accept" recommendation.

Reviewer #2: All comments have been addressed

Reviewer #3: (No Response)

2. Is the manuscript technically sound, and do the data support the conclusions?

Reviewer #2: Yes

Reviewer #3: Yes

3. Has the statistical analysis been performed appropriately and rigorously? 

Reviewer #2: Yes

Reviewer #3: No

4. Have the authors made all data underlying the findings in their manuscript fully available?

Reviewer #2: Yes

Reviewer #3: Yes

5. Is the manuscript presented in an intelligible fashion and written in standard English?

Reviewer #2: Yes

Reviewer #3: Yes

6. Review Comments to the Author

Reviewer #2: I thank the authors for addressing the review comments rigorously. I am happy to recommend publication of this manuscript.

Reviewer #3: The manuscript describes the analysis of a COVID cohort subjected to ICU with regard to various proteins in plasma and endotracheal aspirates and specifically addresses their significance as biomarkers for the course of the infection and the development of fibrosis.

This work examines an important topic. However, important information is missing that needs to be added to make the study comprehensible:

1. the methods section lacks a precise description of how the plasma was purified after blood collection (in EDTA or nitrate tubes). Please add.

2. why were the plasma samples thawed twice before analysis? Please add. Normally plasma is aliquoted and used without thawing for protein analysis, as proteins may be degraded due to multiple thawing cycles.

3. the statistical analysis is not comprehensible to me. As I understand it, the aim was to analyze whether there is a difference in the amount of cytokines in the plasma of different proteins in different groups of the cohort. For example, the amount of CCL2 in patients who survived or died was compared. Why was a logistic regression analysis carried out here and not multiple t-tests with a correction for false discovery rate?

4. A detailed description for how the adjustments regarding age, sex, treatment with dexamethasone, and enrollment APACHE III score were performed, is missing. Please add.

5. Please add the detection limit of the proteins in the electrochemiluminescent immunoassays. Were they around 0.001 pg/ml? Otherwise a negative value of log2=-10 is not possible (TFN-a Figure2).

5. line 353: "Figure 4" is missing

6. Figure S4: CRP, MCP-4 and CXCL12 are missing, please add and remove CCL-13, as this protein is not shown in an other figure/table.

7. PLOS authors have the option to publish the peer review history of their article (what does this mean?). If published, this will include your full peer review and any attached files.

Reviewer #2: **Yes: **Nicholas A Scott

Reviewer #3: No

---

## [Author Response · Author response to Decision Letter 1]

26 Apr 2024

Reviewer 2 General Comments: I thank the authors for addressing the review comments rigorously. I am happy to recommend publication of this manuscript.

Response to Reviewer 2: Thank you for taking the time to extensively review our manuscript and provide your comments, which we feel improved our manuscript. 

Reviewer 3 General Comments: The manuscript describes the analysis of a COVID cohort subjected to ICU with regard to various proteins in plasma and endotracheal aspirates and specifically addresses their significance as biomarkers for the course of the infection and the development of fibrosis.

This work examines an important topic. However, important information is missing that needs to be added to make the study comprehensible.

Response to Reviewer 3: Thank you for thoroughly reviewing our manuscript. We have responded to your comments and made edits to our manuscript/statistical approaches as described below.

Reviewer 3 Comment 1: the methods section lacks a precise description of how the plasma was purified after blood collection (in EDTA or nitrate tubes). Please add

Response to Comment 1: The blood was collected in cell preparation tubes (CPT) containing buffered sodium citrate and a density gradient solution for the isolation of plasma and mononuclear cells simultaneously. The tubes containing whole blood were centrifuged at 1500g for 20 minutes, at which point the tube contained layers of plasma, mononuclear cells and platelets, followed by granulocytes and erythrocytes separated by a gel layer. The plasma was then aliquoted in 0.5 mL aliquots and frozen down, carefully avoiding the mononuclear cell layer. The aliquots were then frozen down and stored at -80C until further aliquoting/analysis. These methods have been added to our methods section in the text (lines 283-289).

Reviewer 3 Comment 2: why were the plasma samples thawed twice before analysis? Please add. Normally plasma is aliquoted and used without thawing for protein analysis, as proteins may be degraded due to multiple thawing cycles.

Response to Comment 2: We recognize that the proteins may be degraded during multiple freeze-thaw cycles, which is why we ensured that the plasma underwent the same number of freeze-thaw cycles prior to analysis. Plasma was processed from whole blood on the same day of sample collection and frozen down (1 freeze). 0.5 mL aliquots were subsequently sub-aliquoted onto plates for our multiplexed analysis (1 thaw followed by 1 freeze). Then, the samples were thawed on the day of analysis (1 thaw), in total resulting in 2 freeze-thaw cycles. This process allowed us to measure several hundred samples simultaneously while minimizing assay batch effects.

Reviewer 3 Comment 3: the statistical analysis is not comprehensible to me. As I understand it, the aim was to analyze whether there is a difference in the amount of cytokines in the plasma of different proteins in different groups of the cohort. For example, the amount of CCL2 in patients who survived or died was compared. Why was a logistic regression analysis carried out here and not multiple t-tests with a correction for false discovery rate?

Response to Comment 3: Our hypothesis was that there would be differences in the amount of cytokines detected in patients who 1) had higher overall in-hospital mortality, 2) fewer ventilator free days, or 3) development of radiographic fibroproliferation. Our pre-determined covariates that could affect cytokine production were age, sex, APACHE III score at the time of enrollment, and treatment with dexamethasone. We used a logistic regression analysis in order to address/control these covariates. 

To demonstrate that the statistical approach does not change our conclusions, please see below a table that shows the p-values based on Mann-Whitney U test for each of our comparisons with the Benjamini-Hochberg correction for false discovery rate. The significantly different cytokines are the same whether the Mann-Whitney test is used, or an unadjusted logistic regression model is used (Supplemental Tables 2-7) A Mann-Whitney test was used rather than a t-test because the cytokine concentrations are not normally distributed. Please see the Response to Reviewers document for statistical tables.

Reviewer 3 Comment 4: A detailed description for how the adjustments regarding age, sex, treatment with dexamethasone, and enrollment APACHE III score were performed, is missing. Please add.

Response to Comment 4: We performed the logistic regression analysis in R studio. Adjustments regarding age, sex, treatment with dexamethasone, and enrollment APACHE III score were performed by including these as covariates in our model. We used the glm function in R with the “binomial” family to specify a logistic regression where the dependent variable was the binary variable of death, fibrosis, or VFDs (high vs low). 

This is also described in the methods section, lines 323-326.

Reviewer 3 Comment 5: Please add the detection limit of the proteins in the electrochemiluminescent immunoassays. Were they around 0.001 pg/ml? Otherwise a negative value of log2=-10 is not possible (TFN-a Figure2).

Response to Comment 5: The lower limit of detection for TNF-a was 0.028 pg/mL. The data point you are referring to was below the lower limit of detection and we have imputed a value of 0.014, resulting in the datapoint shown. This has been clarified in the text (Lines 298-300). We have added the limits of detection for each analyte to Supplemental Table 1.

Reviewer 3 Comment 6: line 353: "Figure 4" is missing

Response to Comment 6: Thank you for catching this error, we have added this in the text (line 365 )

Reviewer 3 Comment 7: Figure S4: CRP, MCP-4 and CXCL12 are missing, please add and remove CCL-13, as this protein is not shown in another figure/table

Response to Comment 7: Thank you for this point. MCP-4 is another name for CCL13. The graph has been adjusted to reflect that label. CXCL12 was not included as the CXCL12 endotracheal aspirate test characteristics did not meet our quality control standards. We did not discuss CRP in the manuscript and it has been removed from all of the tables.

---

## [Editor Report · Decision Letter 2]

17 May 2024

Mediators of monocyte chemotaxis and matrix remodeling are associated with mortality and pulmonary fibroproliferation in patients with severe COVID-19

PONE-D-23-11100R2

Dear Dr. Holton,

We’re pleased to inform you that your manuscript has been judged scientifically suitable for publication and will be formally accepted for publication once it meets all outstanding technical requirements.

Kind regards,

Gernot Zissel, Ph.D.

Academic Editor

PLOS ONE